# Antibacterial Activity of Non-Cytotoxic, Amino Acid-Modified Polycationic Dendrimers against *Pseudomonas aeruginosa* and Other Non-Fermenting Gram-Negative Bacteria

**DOI:** 10.3390/polym12081818

**Published:** 2020-08-13

**Authors:** Anna Maria Schito, Silvana Alfei

**Affiliations:** 1Department of Surgical Sciences and Integrated Diagnostics (DISC), University of Genoa, Viale Benedetto XV, 6, I-16132 Genova, Italy; amschito@unige.it; 2Department of Pharmacy (DiFAR), University of Genoa, Viale Cembrano 4, I-16148 Genova, Italy

**Keywords:** novel bactericidal agents, polycationic dendrimers, *Pseudomonas aeruginosa*, antibiotic-resistant, non-fermenting pathogens

## Abstract

Due to the rapid increase of antimicrobial resistance with ensuring therapeutic failures, the purpose of this study was to identify novel synthetic molecules as alternatives to conventional available, but presently ineffective antibiotics. Variously structured cationic dendrimers previously reported have provided promising outcomes. However, the problem of their cytotoxicity towards eukaryotic cells has not been completely overcome. We have now investigated the antibacterial activities of three not cytotoxic cationic dendrimers (G5Ds: G5H, G5K, and G5HK) against several multidrug-resistant (MDR) clinical strains. All G5Ds displayed remarkable activity against MDR non-fermenting Gram-negative species such as *P. aeruginosa*, *S. maltophilia*, and *A. baumannii* (MICs = 0.5–33.2 µM). In particular, very low MIC values (0.5–2.1 µM) were observed for G5K, which proved to be more active than the potent colistin (2.1 versus 3.19 µM) against *P. aeruginosa*. Concerning its mechanism of action, in time-killing and turbidimetric studies, G5K displayed a rapid non-lytic bactericidal activity. Considering the absence of cytotoxicity of these new compounds and their potency, comparable or even higher than that provided by the dendrimers previously reported, G5Ds may be proposed as promising novel antibacterial agents capable of overcoming the alarming resistance rates of several nosocomial non-fermenting Gram-negative pathogens.

## 1. Introduction

The rapid and worldwide increase in antimicrobial resistance among bacterial pathogens, dramatically associated with therapeutic failures, has become a global concern urgently requiring alternative curative options [1].

This situation particularly applies to nosocomial pathogens that are responsible for a number of severe clinical conditions in immunocompromised and critically ill individuals [1]. Aerobic non-fermenting Gram-negative bacilli such as *Acinetobacter baumannii*, *Pseudomonas aeruginosa*, and *Stenotrophomonas maltophilia* are emerging as clinically relevant superbugs, contributing significantly, with their worrying resistance levels, to numerous therapeutic failures [2].

These species, in addition to their intrinsic resistance mechanisms, are rapidly becoming multidrug- or even pan-drug-resistant to most life-saving drugs, thus making the search for new antimicrobial agents urgent. An appealing antibacterial molecule that has been recently reconsidered as a therapeutic agent, particularly active against multidrug-resistant Gram-negative pathogens, is colistin (polymyxin E).

Colistin, like other polymyxins, is a natural cationic antimicrobial cyclic peptide belonging to the class of antimicrobial cationic peptides (CAMPs) [3,4,5,6], able to display a remarkable potency against the non-fermenting species, usually causing nosocomial serious infections, such as *P. aeruginosa*, *Acinetobacter* spp., and *S. maltophilia.* Unfortunately, its widespread use is critically thwarted by its well-known nephro- and neurotoxicity [7].

More generically, CAMPs are antimicrobial agents with a broad spectrum of action, active on a wide variety of Gram-positive and Gram-negative bacteria, fungi, protozoa, and yeast [8,9].

According to several observations, CAMPs may damage and kill microorganisms by interfering with several specific bacterial vital processes [10], but CAMPs are mainly reported as membrane-active compounds and membrane disruptors [8,9,10,11].

Briefly, thanks to their cationic structure associated with *N*-terminal fatty acids tails, CAMPs interact electrostatically with the anionic constituents of bacterial cell membranes, such as lipopolysaccharides (LPS) and phospholipids, and diffuse inside them.

These phenomena cause alterations in the integrity of the membranes, pore formation, and permeabilization of the cell wall that ultimately lead to loss of bacterial cytoplasmic content and cell death [8,9,10,11].

Despite their efficacy, rapid action, and a low incidence in developing resistance [12], due to their mechanism of action, CAMPs are endowed with low biocompatibility and high toxicity against eukaryotic cells [8].

In recent years, the research trends have been focused on developing less toxic synthetic mimics of CAMPs, including cationic peptides, positively charged polymers, and polycationic dendrimers [8].

Dendrimers represent a privileged class of molecules completely different from linear polymers, extensively studied for biomedical applications [13,14,15,16].

In the last years, several polycationic dendrimers, such as PEIs dendrimers [17,18], polypropylene imine (PPI) [19], polyamidoamine (PAMAMs) [20,21], or peptide-based dendrimers [22,23,24,25], have been developed [26] for the treatment of infections sustained by multidrug-resistant bacteria.

Unfortunately, their clinical application is hampered by several issues, such as low biodegradability, susceptibility to opsonization, high level of toxicity to mammalian cells, including hemolytic toxicity, cytotoxicity and hematological toxicity, and fast clearance [13,14,15,16].

In this regard, polyester-based dendrimer scaffolds, peripherally functionalized with natural amino acids, might also represent very appealing materials because of their good biodegradability [27,28,29].

On this background, three fifth-generation polyester-based dendrimers (G5Ds) with a biodegradable inner matrix (G5) and a surface decorated with amino acids (Scheme 1) were selected among a library of 15 polycationic *homo*- and *hetero*-dendrimers [30] and have been tested as potential antimicrobials against a vast array of Gram-positive and Gram-negative human pathogens.

In particular, G5Ds encompass a tri-hydroxyl *core*, namely 2,2-*bis*-hydroxymethylpropanol, repeated units of the AB2 monomer, and 96 peripheral hydroxyls groups esterified with *L*-lysine (G5K), *L*-histidine (G5H), or a mixture 46/50 of *L*-histidine and *L*-lysine (G5HK), respectively (Scheme 1) [30].

## 2. Materials and Methods

### 2.1. Microorganisms

A total of 36 pathogens were used in this study as better specified in Appendix A of SM: 35 were clinical strains isolated from human specimens (S. Martino Polyclinic Hospital, Genoa, Italy) and identified by VITEK^®^ 2 (Biomerieux, Firenze, Italy) or matrix-assisted laser desorption/ionization-time-of-flight (MALDI-TOF) mass spectrometric technique (Biomerieux, Firenze, Italy). Among the two non clinical strains, one was a strain of *Pseudomonas straminea* isolated from flowers studied in the Interreg ALCOTRA project; “ANTEA 1139” was also included and one was an ATCC strain (*Pseudomonas aeruginosa* ATCC 27853, American Type Culture Collection, Manassas, VA, USA) (Table 1, Section 3.2). Of the tested organisms, eight were Gram-positive species, including one *Enterococcus faecalis* resistant to vancomycin (VAN-R) and one susceptible (VAN-S), one *Enterococcus faecium* VAN-R and one VAN-S, one methicillin-resistant *Staphylococcus aureus* (MRSA) and one susceptible, one methicillin-resistant *Staphylococcus epidermidis* (MRSE) and one susceptible. Out of the remaining clinical strains, 26 belonged to Gram-negative species, including two *Escherichia coli*, two *Klebsiella pneumoniae*, one *Proteus mirabilis*, 10 *Pseudomonas aeruginosa*, one *Pseudomonas putida*, one *Pseudomonas fluorescens*, four *Stenotrophomonas maltophilia*, four *Acinetobacter baumannii*, and one *Acinetobacter pittii.* The antibiotic resistance patterns of the pathogens tested against the three dendrimers were obtained by VITEK^®^ 2 or Kirby-Bauer/e-test techniques when necessary and are detailed in the SM (Appendix A).

### 2.2. Antimicrobial Assays

The minimal inhibitory concentrations (MICs) of the three G5Ds (G5K, G5H, and G5HK) on the 36 pathogens were determined following the microdilution procedure detailed by the European Committee on Antimicrobial Susceptibility Testing EUCAST [31].

Briefly, overnight cultures of bacteria were diluted to yield a standardized inoculum of 1.5 × 10^8^ CFU/mL.

Aliquots of each suspension were added to 96-well microplates containing the same volumes of serial twofold dilutions (ranging from 0.015 to 256 μg/mL) of each dendrimer to yield a final concentration of about 5 × 10^5^ cells/mL. The plates were then incubated at 37 °C. After 24 h of incubation at 37 °C, the lowest concentration of dendrimer that prevented a visible growth was recorded as the MIC. All MICs were obtained in triplicate, the degree of concordance was in all the experiments 3/3, and standard deviation (±SD) was zero.

### 2.3. Killing Curves

Killing curve assays for G5K were performed on three representative isolates of *P. aeruginosa* (strains 230, 249, and 259) all susceptible to colistin, including one, strain 249, mucous, two representative isolates of *S. maltophilia* (strain 11 susceptible to trimethoprim/sulfamethoxazole and strain 16 intermediate), and two representative strains of *A. baumannii* (strains 24 and 25) as previously reported [32]. Experiments were performed over 24 h at G5K concentrations of four times the MIC for all strains.

A mid logarithmic phase culture was diluted in MH broth (10 mL) containing 4 × MIC of the selected compound in order to give a final inoculum of 1.0 × 10^5^ CFU/mL. The same inoculum was added to cation-supplemented Mueller–Hinton broth (CSMHB) (Merck, Darmstadt, Germany), as a growth control.

Tubes were incubated at 37 °C with constant shaking for 24 h. Samples of 0.20 mL from each tube were removed at 0, 2, 4, 8, and 24 h, diluted appropriately with a 0.9% sodium chloride solution in order to avoid carryover of the dendrimer being tested, plated onto MH plates, and incubated for 24 h at 37 °C. Growth controls were run in parallel.

The percentage of surviving bacterial cells was determined for each sampling time by comparing colony counts with those of standard dilutions of the growth control. Results have been expressed as log10 of viable cell numbers (CFU/mL) of surviving bacterial cells over a 24 h period.

Bactericidal effect was defined as a 3 log10 decrease of CFU/mL (99.9% killing) of the initial inoculum. All time-kill curve experiments were performed in triplicate.

### 2.4. Evaluation of the Antimicrobial Effect of G5K by Turbidimetric Studies

The study of the antimicrobial activity of G5K was carried out measuring the optical density variations (OD) as a function of time in cultures of the same strains employed for the time-killing experiments (three trains of *P. aeruginosa*, two of *A. baumannii*, and two of *S. maltophilia*) at a wavelength of 600 nm in a Thermospectronic spectrophotometer (Ultrospec 2100pro, Amersham Biosciences, Little Chalfont, UK) [33].

Bacterial cells were harvested from 10 mL of bacterial cultures in MH broth, and cell number was adjusted in order to produce a heavy inoculum (OD adjusted to 0.2) corresponding to 10^8^ cells/mL. Cell suspensions were treated with or without G5K at concentration equal to 4 × MIC and incubated at 37 °C. After 30 min and 1, 2, 3, 4, 5, 6, and 24 h of incubation, aliquots were taken from the cultures, and absorbance values were recorded at 600 nm.

Measurements were blanked with MH broth containing an equivalent amount of G5K being tested. Experiments were performed in triplicate. The number of CFU was determined in parallel as described in the time-killing section and compared with the untreated sample.

## 3. Results and Discussion

### 3.1. Selection of Positively Charged Dendrimers G5K, G5H, and G5HK

To select the most promising dendrimers (G5Ds) in the library of 15 available compounds, a screening was made, based on their polymer structure, their high generation, their high density of cationic charges, due to the presence of several natural *L*-amino acids on surface, their uncharged inner architecture, the type of amino acids, and their absence of cytotoxicity versus eukaryotic cells [30].

Compared to small drug molecules, polymer structures own several advantages such as more long-term activity, limited residual toxicity, chemical stability and non-volatility [8,34].

The high generation assures high multivalence, which has been reported to be a pivotal feature for an effective antimicrobial activity of cationic dendrimers [26].

High multivalence translates to high density of positive charge that promotes electrostatic interactions with bacterial membranes, which is the first step of the mechanism of action of cationic antimicrobial devices [8,34,35].

The presence of several *L*-amino acids, as natural molecules to confer the essential polycationic character, harmonized by the not charged macromolecular inner matrix, would provide the typical amphiphilic asset of synthetic antimicrobial peptides [8,36] and peptide dendrimers [22,23,24,25], which proved antimicrobial activity and selectivity for bacteria, superior to those of native CAMPs, and limited or even absent toxicity towards mammalian cells [8,22,23,24,25,36].

Finally, those dendrimers containing *L*-lysine and *L*-histidine were preferred, since cationic polymers and/or peptides containing lysine [8,36] or opportunely modified imidazole groups [37] showed considerable antibacterial effects and low hemolytic toxicity.

### 3.2. Antimicrobial Activities of G5K, G5H, and G5HK

MIC values for all three dendrimers were obtained analyzing a total of 36 strains, the majority being of clinical origin. G5K, G5H, and G5HK displayed extremely high MIC values against Gram-positive isolates such as *S. aureus*, including a methicillin-resistant strain (MRSA), *S. epidermidis*, including a methicillin-resistant strain (MRSE), *E. faecalis*, including a vancomycin-resistant strain (VRE), and *E. faecium*, including a VRE strain.

Representative strains of *Enterobacteriaceae* such as *E. coli*, *K. pnemoniae*, and *P. mirabilis* displayed the same pattern, providing MICs values > 0.0329–0.0334 mM, and therefore were not further investigated. These strains were considered not susceptible to G5Ds, even if in a previous study [38], a second-generation polyester-based alanine-modified cationic dendrimer, similar to G5Ds, was considered active against *E. coli* when showed a MIC value of 0.100 mM, i.e., about threefold higher than those here observed for all the G5Ds [38].

Interestingly, the three compounds manifested consistent inhibitory activities against non-fermenting Gram-negative pathogens, including *P. aeruginosa*, *S. maltophilia*, and *A. baumannii* (Table 1).

G5K showed to be the most powerful dendrimer against these species, in terms of MIC values, followed by G5HK and G5H. MIC values against the susceptible species were quite uniform for all the three dendrimers, particularly against *P. aeruginosa*, and were totally independent of their various antibiotic resistance patterns, exception made for colistin.

The activity of G5K against susceptible *P. aeruginosa* was only slightly lower than that showed by a strongly active peptide dendrimer containing lysine and leucine (3GKL) previously synthetized, but far higher (3.6-fold) than that of another dendrimer peptide bH1 reported in the same study [39].

In general, G5H and G5HK were less potent than both 3GKL and bH1, but against some species of *P. aeruginosa*, G5HK displayed an antimicrobial effect comparable or even higher than that showed by bH1 against *P. aeruginosa* PAO1 [39].

The antimicrobial activity of G5K against *A. baumannii* was comparable to that of G3KL, and 3.2–6 times higher than that of bH1 [39].

Moreover, a multidrug-resistant strain (259), refractory even to ceftazidime/avibactam, was susceptible to the inhibitory activity of G5K, with a MIC value of 64 mg/L, corresponding to 0.00207 mM, 2.07 µM.

In this regard, the activity of G5K was in the range of the MICs observed for 3GKL, which proved MIC values of 0.9–7.0 µM, and far more active than bH1, which provided a MIC value > 26.7 µM [39].

Both G5H and G5HK were less active than 3GKL, but if compared to bH1, G5H displayed similar activity (33.2 µM versus >26.7 µM), while G5HK was 1.6-fold more effective [39].

Moreover, although breakpoints for colistin against *P. aeruginosa* are slightly discordant between the EUCAST and the CLSI committees, (i.e., ≤2 µg/mL for susceptibility and >2 µg/mL for resistance according to EUCAST and ≤2 µg/mL for susceptibility and ≥4 µg/mL for resistance for CLSI), EUCAST identifies the concentration of 4 µg/mL for *P. aeruginosa* as a “problematic” concentration, falling in an “area of technical uncertainty”.

Interestingly, this last value corresponding to 3.19 µM is higher than the MIC of G5K observed here on *P. aeruginosa* (2.1 μM), thus implying that G5K might be considered more potent on this pathogen than colistin.

Interestingly, among the *Pseudomonas* genus, *P. aeruginosa* showed to be the less susceptible species to the three dendrimers, while other species, such as *P. putida*, and *P. fluorescens*, that are widely present in nature and that occasionally can behave as opportunistic pathogens of humans, appeared to be far more susceptible (Table 1).

In this regard, MIC values of the three dendrimers reported for *P. putida* and *P. fluorescens* closely resemble those obtained on *P. straminea*, a typical endophytic bacterium, endowed with antagonistic activity against fungal pathogens. A similar increase in potency of G5HK and G5H, in terms of MIC values, was observed against *A. pittii* compared to *A. baumannii* (Table 1).

*P. aeruginosa* strain 265, resistant to colistin, was refractory to the activity of the three dendrimers confirming that, as for other CAMPs [6,7], and also for dendrimers, the initial interactions between the positive charge of the matrices with the negatively charged bacterial surfaces seems to be crucial for their activity.

Except for this case, the antimicrobial activity of G5K against all strains of *P. aeruginosa* essayed (MICs = 2.1 µM) was 6.5-fold higher than that showed by the most active peptide dendrimer synthetized previously by Niederhafner et al. and tested against a not characterized strain of *P. aeruginosa* (MICs = 13.8 µM) [40].

Concerning this antimicrobial peptide dendrimer, except for multidrug-resistant strain 259 and strain 253, G5HK also proved to be far more active, providing MICs values by 1.6–3.3 lower [40].

The difference in antimicrobial potencies among the three dendrimers, in terms of intrinsic potency, may be ascribed to several differences in their molecular structure. In fact, although the three dendrimers have the same number of cationic groups, conferred by the presence of amino acids on the surface, they differ in molecular weight, in the nature of peripheral amino acid, and therefore in the type of cationic groups, which are, among others, pivotal factors strongly influencing the antimicrobial activity of cationic polymers [8].

In this regard, the higher antimicrobial activity of G5K associated to its previously demonstrated low toxicity against mammalian cells [30] may be ascribed to the presence of *L*-lysine (K), as natural amino acid responsible for the cationic character of this particular dendrimer.

As confirmation to this assumption, it has been reported that the natural cationic polymers known as poly(*ε*-lysine) showed a substantial antimicrobial activity against Gram-negative pathogens, including different strains of *E. coli*, *P. fluorescens*, *P. putida*, *P. aeruginosa*, *Serratia marcescens*, and *Salmonella typhimurium*, associated to a high biocompatibility, biodegradability, and low toxicity, both in vitro and in vivo essays [8].

In addition, a synthetic CAMP, presenting a high content in K and characterized by the sequence KKKKKKAAXAAWAAXAA-NH_2_, proved to possess a high selectivity for bacterial membranes, combined with high activity toward a wide spectrum of Gram-negative and Gram-positive bacteria and yeast [36]. Finally, long hydrophilic cationic mutants of hydrophobic synthetic CAMPs not bearing quaternary primary ammonium groups, as K, and consequently G5K, proved to be membrane active materials against bacteria but showed strikingly reduced hemolytic toxicity and drastically enhanced selectivity [8,37,41].

While the synthetic agent, with sequence KKKKKKAAXAAWAAXAA-NH_2_ [36] containing lysine, showed a wide antimicrobial activity, G5K manifested a much more selective potency for non-fermenting Gram-negative bacteria and was demonstrated inactive on several enterobacterial and Gram-positive species.

In this regard, it has to be considered that, in the structure of the synthetic CAMP, the K residues are linked together or with other amino acids, with a peptide bond, and therefore, only the *ε* amino groups of K, and the terminal amino group of the peptide provide the cationic character responsible for the interactions with bacteria membranes and for the bactericidal activity.

On the contrary, in the structure of G4K, all the amino groups of the 96 residues of K, i.e., 192 NH_2_, greatly contribute to the strong cationic character of the dendrimer.

It follows that G5K will have a much greater affinity for interacting with the membranes of Gram-negative bacteria than with that of Gram-positive ones, where the anionic character is weaker.

In addition, compared to polymyxins, the three dendrimers analyzed here, and particularly G5K, showed a more specific and narrower spectrum of action against the Gram-negative families, being, different from poly(*ε*-lysine) [8], highly active only against the non-fermenting species.

Sparing a large part of the physiologic microbiome, since the Gram-positives and the enterobacteria are not inhibited, may render the usage of G5K more attractive when treating severe infections, provided its pharmacokinetic, pharmacodynamic, and safety characteristics are found to be favorable. This important and new feature may probably be ascribed to the specific chemical structure of the LPS or the outer membrane (OM) of the microorganisms not included in the G5K spectrum of action. In this regard, it was reported that important differences in LPS exist between *E. coli* strains and *P. aeruginosa* strains [42,43].

### 3.3. Time-Killing Curves

Time kill experiments were performed with the most powerful dendrimer, i.e., G5K at concentrations equal to 4 × MIC on three strains of *P. aeruginosa*, two strains of *A. baumannii*, and one of *S. maltophilia*. As depicted in Figure 1, showing the most representative curves obtained for each species, G5K possessed the strongest bactericidal effect against *P. aeruginosa*, since a rapid decrease of >4 logs in the original cell number was evident already after one hour of exposure and was maintained for at least 6 h after incubation.

Against *A. baumannii*, G5K proved to be bactericidal, by inducing a 3 log decrease of the starting inoculum after 6 h of exposure, while the inhibition of this dendrimer against *S. matophilia* was less pronounced (2 log) after the same time.

Regrowth was noted after 24 h of incubation with G5K for all the three species tested. This behavior is similar to that already observed for cationic bactericidal peptides that kill on contact, as colistin, where the initial killing is rapid, being produced as soon as 5 min after antibiotic exposure, and is followed by regrowth after 24 h [6].

### 3.4. Effect of G5K on the Growth Curve of P. aeruginosa, A. baumannii, and S. maltophilia

The kinetics of growth in MH broth in the absence or presence of G5K at a concentration of 4 × MIC was followed at 600 nm for a period of 6 h on selected strains of *P. aeruginosa*, *A. baumannii*, and *S. maltophilia*. Figure 2 shows the results obtained on one representative strain of *P. aeruginosa.* While, as expected, the control culture showed a logarithmic turbidimetric increase, the presence of G5K resulted in a complete inhibition of growth not followed, during time, by a decrease of the optical density suggesting that, despite the compound being bactericidal against the pathogen, as confirmed in the previously reported experiments, it is unable to induce frank cell lysis. Similar results were obtained for all the strains of the three species analyzed.

Concerning the mechanism of action of G5K, it is possible to advance the hypothesis that, like other CAMPs, G5K might be electrostatically attracted to the highly anionic OM of *P. aeruginosa* and of all the other susceptible species. Thereafter, binding may cause the observed effect, possibly through displacement of Ca^+^ and Mg^+^ ions, destabilization of LPS, pores formation, and diffusion towards CM and electrostatic absorption as well. Thereafter, CM depolarization and increasing permeabilization by further pores formation could lead to cell death by several mechanisms, including loss of membrane potential, inhibition of biosynthetic pathways involving ATP, DNA, RNA or proteins, free radical production, and leakage of the cytoplasmic content including crucial inorganic species as K^+^, Cs^+^, Na^+^, Li^+^, and phosphate [8,44]. The inactivity of G5K on the colistin-resistant *P. aeruginosa* strain included in our study confirms the above-mentioned hypotheses.

It should be remembered that the lytic mechanism attributed to the currently utilized membrane-damaging antimicrobials, such as colistin, although still controversial, resides in the presence of a *N*-terminated hydrophobic fatty acid side chain that, added to the positively charged peptide ring, confers to the molecule the amphiphilic character necessary for allowing its diffusion through the OM and towards the CM, pivotal for the disruption of the CM bilayer [6,8].

The absence of a similar fatty acid side chain in G5K, by enhancing its hydrophilic character, may limit its diffusion towards CM causing its disruption, and may justify the fact that, while being bactericidal probably inducing membrane impairments such as depolarization and destabilization, it lacks both lytic properties on bacteria and cytotoxic action versus mammalian cells [8,42,43].

Anyway, the high capacity of G5K to adhere to negative phosphate groups of genetic material and membrane phospholipids and the inability of spreading through the lipid bilayers of humans cells in a detrimental way, causing membrane disruption and cells lysis, had already been shown [30].

Images displaying such findings are available in SM (Appendix A). As shown in Appendix A, compound G5H, possessing the lowest antibacterial activity, was the only compound tested that was unable to bind phosphate groups of genetic materials, thus confirming that the bactericidal effects of G5Ds is mainly based on their electrostatic interaction with bacterial OM.

As for our knowledge, only another study has been reported in the recent literature concerning the usage of biodegradable amino acids-modified polyester-based dendrimers as antimicrobial agents with limited toxicity [38]. The authors, probably inspired by a study previously reported [30], using the same monomer, *bis*-HMPA, have synthesized several scaffolds from first to fifth generation, made cationic by peripheral esterification with the unnatural amino acid β-alanine [38]. Though fast biodegradation of the dendrimers was reported, a non-completely characterized antimicrobial activity against *E. coli* K12, associated with an acceptable toxicity against human dermal fibroblasts and mouse monocyte cells, was shown. In this regard, significant antibacterial effects were observed only for the G2 dendrimers at a very high dosage (100 µM) [38]. These authors did not investigate the possible mechanisms of action of the dendrimer against *E. coli*, but speculated that G2 resulted inhibitory, due to the efficacious combination of charge density and mobility, promoting a lethal interaction with the bacterial membrane [38].

## 4. Conclusions

The scope of the present work was to identify novel and unconventional antimicrobial agents as promising alternatives to traditional antibiotics, often ineffective against MDR bacteria.

Therefore, following the current trend that considers cationic dendrimers as suitable candidates, three fifth-generation dendrimers (G5Ds) containing lysine, histidine, or a mixture of both (G5K, G5H and G5HK) have been herein investigated.

The antibacterial activity of G5K, G5H, and G5HK against a variety of strains of Gram-positive and Gram-negative bacteria was assessed by determining the MIC values by standard microdilution methods.

The bactericidal potency and the mechanism of action of the most active compound (G5K) was investigated by time-killing experiments and turbidimetric essay, rarely reported in studies concerning antimicrobial dendrimers, on clinical strains of *P. aeruginosa*, *A. baumannii*, and of *S. maltophilia*. All G5Ds displayed remarkable activity against all the strains of non-fermenting Gram-negative bacteria just mentioned.

In particular, G5K showed a very strong antibacterial activity, higher than that of potent colistin and of other antimicrobial cationic dendrimers endowed with limited cytotoxicity previously reported.

Moreover, G5K displayed a rapid bactericidal activity against *P. aeruginosa*, *A. baumannii*, and *S. maltophilia*, which is promising for a low tendency of this novel antimicrobial agent to develop resistance, and, on *P. aeruginosa*, a non-lytic mechanism of action.

G5Ds and mainly G5K represent the first examples of nontoxic, highly biodegradable, polyester-based amino acids-modified dendrimers that are actually endowed with remarkable antimicrobial activity. They are worthy to be regarded as promising alternatives to several commonly used antibiotics now plagued by the burden of widespread resistance. These compounds could be viewed as novel and less toxic replacements of CAMPs (including colistin) or previously prepared cationic polymers, particularly for the treatment of infections sustained by the non-fermenting Gram-negative pathogens. Moreover, G5K represents a novel template molecule for the development of new antimicrobials with a broader spectrum of action by structural modification of the inner matrix.

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
