# Peer review of "Antibacterial Activity of Non-Cytotoxic, Amino Acid-Modified Polycationic Dendrimers against Pseudomonas aeruginosa and Other Non-Fermenting Gram-Negative Bacteria"

_polymers, 2020, doi:10.3390/polym12081818_

Round 1

Reviewer 1 Report

The manuscript polymers-880339 entitled “Antibacterial activity of non-cytotoxic, amino acid-modified polycationic dendrimers against Pseudomonas aeruginosa and other non-fermenting Gram-negative bacteria" reported the antibacterial activity of some amino-acids against several Gram-negative bacteria strains. The topic is of high current interest. However, some technical concerns must be raised.

GENERAL COMMENTS

In its current state, the level of English throughout your manuscript does not meet the journal's desired standard. There are some grammatical and spelling errors and full stops missing as well as instances of badly worded/constructed sentences. Please check the manuscript and refine the language carefully. I suggest that you should ask several colleagues who are skilled authors to check the English before your submission.

The main content of the abstract should include the briefly purpose of the research, the principal result and major conclusion. The abstract, in the present form, is not adequate. Additionally, (i) the authors must state the revise justification in the abstract to support the study; (ii) some results about antibacterial activity should be nice. (iii) Conclusion in the abstract is too general, authors shall state the important finding in this study.

The introduction section is too long. Additionally, there is too many general information about irrelevant topics. Authors must summarized those general information and improve this section with relevant information. In the same way, a deep review of pertinent literature is necessary.

In the current form, discussion section is more like general reporting. Authors spend too much space for general information. However, authors must focus on discuss and compare their finding with previous report on this field.  This section does not explain what was done on antibacterial properties of bio-compounds analyzed. Additionally, a deep review of pertinent literature was not used.

Why a statistical analysis was not performed?

The conclusion section is redundant as results already summarized and analyzed. In the present form this section is not adequate. This section must be improved.

Author Response

Comments and Suggestions for Authors

The manuscript polymers-880339 entitled “Antibacterial activity of non-cytotoxic, amino acid-modified polycationic dendrimers against Pseudomonas aeruginosa and other non-fermenting Gram-negative bacteria" reported the antibacterial activity of some amino-acids against several Gram-negative bacteria strains. The topic is of high current interest. However, some technical concerns must be raised.

GENERAL COMMENTS

In its current state, the level of English throughout your manuscript does not meet the journal's desired standard. There are some grammatical and spelling errors and full stops missing as well as instances of badly worded/constructed sentences. Please check the manuscript and refine the language carefully. I suggest that you should ask several colleagues who are skilled authors to check the English before your submission.

According to the Reviewer, the manuscript has been carefully checked in order to improve and refine English by removing grammatical and spelling errors, adding full stops where missing and enhancing the construct of sentences. In order to perform successfully the language revision, mother tongue professor Deirdre Kantz, (University of Genoa and Pavia) has been involved. The authors thank the professor for her contribute.  

The main content of the abstract should include the briefly purpose of the research, the principal result and major conclusion. The abstract, in the present form, is not adequate. Additionally, (i) the authors must state the revise justification in the abstract to support the study; (ii) some results about antibacterial activity should be nice. (iii) Conclusion in the abstract is too general, authors shall state the important finding in this study.

In order to satisfy the Reviewer request the Abstract has been extensively modified The revised abstract now includes the purpose of the research (lines 13-16), the principal results (lines 29-35) and major conclusions (lines 36-40). The authors are now confident that the revised abstract is adequate, since it encompasses the justifications to support the study (lines 17-19), the results about the antibacterial activity of dendrimers under study in terms of MIC values and time killing experiments (lines 31, 32, 33 and 34-35) and specific clear conclusions (lines 36-40).

The introduction section is too long. Additionally, there is too many general information about irrelevant topics. Authors must summarized those general information and improve this section with relevant information. In the same way, a deep review of pertinent literature is necessary.

According to the Reviewer request, the introduction section has been shortened. Please reconsider the revised version of the Introduction section. Ref. 16 and 17 (unrevised manuscript) have been removed. General information has been summarized, while relevant information associated to pertinent literature have been included. Please see new references: Ref. 9, 12, 17-25 in the references list and at lines 66, 83 and 94-96. Furthermore, in accordance with a request from the other Reviewer, a new relevant Chart (Chart 1) has been included in the revised manuscript. Please see Chart 1 and lines 114-118 and 120-121.

In the current form, discussion section is more like general reporting. Authors spend too much space for general information. However, authors must focus on discuss and compare their finding with previous report on this field.  This section does not explain what was done on antibacterial properties of bio-compounds analyzed. Additionally, a deep review of pertinent literature was not used.

As requested, Discussion section has been extensively modified, focusing on the discussion of the results on the G5Ds and reporting comparisons with previous reports in this field. Moreover, a more deep review of pertinent literature, associated with the reported case studies have been included. Please see lines 245-250 and Ref. 38, 262-269 and 273-276 (Ref. 39), 277-284 and 298-303 and Ref. 40.

Concerning the criticism raised by the Reviewer regarding the investigations performed to evaluate the antibacterial properties of analyzed dendrimers (in this regard the authors precise to the Reviewer that the analyzed molecules are not bio-compounds but synthetic compounds), the authors are confident that the entire discussion now clearly details the antimicrobial properties of the three analyzed dendrimers.

MIC values obtained have been presented, discussed and compared to those of antibiotics significant for the study (colistin) and of antimicrobial dendrimers previously reported.

The difference in activity of the three dendrimers has been discussed and explained. The bactericidal behavior of the most active dendrimer (G5K) has been investigated by time-killing experiments, rarely performed in studies the field of antibacterial polymers, which have been extensively discussed and explained. Moreover, also the non-lytic mechanism of killing, observed performing turbidimetric studies, different from the lytic one frequently reported for analogous antibacterial agents, has been discussed and explained. Please reconsider with more attention the discussion section of the revised paper. 

Why a statistical analysis was not performed ?

Since all the experiments made in triplicated to determine the MIC values showed degrees of concordance of 3/3, there was no need of statistical analysis. Standard deviation (SD) has not been reported because equal to zero. Furthermore, according to the Analytical Chemistry, since the distribution of MICs, follows a trend defined “probably not normal”, the SD would not be anyway appropriate as a descriptive parameter.

However, for greater clarity, sentences showing the degree of concordance of our experiments and the SD values (zero) have been reported in the Materials and Methods section and in the footnote of Table 1.  Please see lines 179-180 and 257, respectively.

In this contest, it is difficult or impossible to find in literature works in which a statistical analysis of MICs or MBCs has been performed. Differently, MIC values are always reported without standard deviations. 

Please see as examples:

Scorciapino, M.A.; Pirri, G.; Vargiu, A.V.; Ruggerone, P.; Giuliani, A.; Casu, M.; Buerck, J.; Wadhwani, P.; Ulrich, A.S.; Rinaldi, A.C. A novel dendrimeric peptide with antimicrobial properties: Structure-function analysis of SB056. Biophysical Journal 2012, 102, 1039–1048, doi:10.1016/j.bpj.2012.01.048 (Supporting Information).

Stach, M.; Siriwardena, T.N.; Köhler, T.; Van Delden, C.; Darbre, T.; Reymond, J.-L. Combining topology and sequence design for the discovery of potent antimicrobial peptide dendrimers against multidrug-resistant pseudomonas aeruginosa. Angewandte Chemie - International Edition 2014, 53, 12827–12831, doi:10.1002/anie.201409270.

Hemolytic and Antimicrobial Activities of a Series of Cationic Amphiphilic Copolymers Comprised of Same Centered Comonomers with Thiazole Moieties and Polyethylene Glycol Derivatives

by R. Cuervo-Rodríguez ,A. Muñoz-Bonilla ,F. López-Fabal andM. Fernández-García

Polymers 2020, 12(4), 972; https://doi.org/10.3390/polym12040972.

Similarly, when time killing curves are reported, only the most representative EXPERIMENT without error bars is reported in graph.

Please see: Michael J. Pucci, Steven D. Podos, Jane A. Thanassi, Melissa J. Leggio, Barton J. Bradbury, and Milind Deshpande. In Vitro and In Vivo Profiles of ACH-702, an Isothiazoloquinolone, against Bacterial Pathogens. ANTIMICROBIAL AGENTS AND CHEMOTHERAPY, 2011, 55, 2860–2871. doi:10.1128/AAC.01666-10.

The conclusion section is redundant as results already summarized and analyzed. In the present form this section is not adequate. This section must be improved.

Conclusion section has been improved by highlighting what was the scope of the work, the microbiologic investigations performed, the relevance of the results, compared with previous findings and/or with the activity of one of the most active available antibiotic in case of resistance, as colistin, and finally the novelty and originality of the study.

Reviewer 2 Report

Recommendation: Publish after major revisions noted.

Comments:

This manuscript by Schito studied and proved the antibacterial activities of three reported dendrimers modified with lysine and/or histidine. The work on antibacterial tests of known compounds appears to have not been reported before. The following items should be addressed prior to publication:

  1. In the manuscript, a chart including the core structure G5 and the modifications of the three G5Ds are needed to visualize the compounds tested for their antibacterial activities. Also, cite Ref 21 at the chart.

  1. At Line 48, replace "this compound" with Colistin. Try to avoid using this, it, they and

  1. Section 2.1 was not needed as well as all the NMR data in SM, since the compounds were reported in ref 21 from the same authors. Nevertheless, the authors needed to prepare the compounds in the identical way reported in Ref 21 and were sure the purity was good. The description of G5K, G5H and G5HK should be moved to Introduction.

  1. The sentences at line 96-98 concluding the toxicities, if newly discovered, should be moved to Results section. But Ref 21 was also cited here, does that mean the results in the Supplementary Materials and the Table S3 and S4 were reported in the Ref already? If so these results are not needed to be repeated in SM again (kind of self-plagiarism?)

  1. Keep the references with same formats. For example, capital or don’t capital all the first letters at titles of the references.

Author Response

This manuscript by Schito studied and proved the antibacterial activities of three reported dendrimers modified with lysine and/or histidine. The work on antibacterial tests of known compounds appears to have not been reported before. The following items should be addressed prior to publication:

  1. In the manuscript, a chart including the core structure G5 and the modifications of the three G5Ds are needed to visualize the compounds tested for their antibacterial activities. Also, cite Ref 21 at the chart.

As requested by the Reviewer a chart (Chart 1) including the structure of G5 and the peripheral modifications that lead to the structures of the three dendrimers tested for their antimicrobial activity has been inserted in Introduction section after line 118 at page 4. Ref 21 of the original manuscript now Ref. 30 has been cited in the Chart 1 caption (line 121).

  1. At Line 48, replace "this compound" with Colistin. Try to avoid using this, it, they and

The authors think that the sentence of the Reviewer is not finished. Anyway “this compound” has been replaced with “colistin” as requested (line 58 of the revised manuscript).

  1. Section 2.1 was not needed as well as all the NMR data in SM, since the compounds were reported in ref 21 from the same authors. Nevertheless, the authors needed to prepare the compounds in the identical way reported in Ref 21 and were sure the purity was good. The description of G5K, G5H and G5HK should be moved to Introduction.

As suggested by the Reviewer, Section 2.1 has been removed (lines 147-151), as well as the NMR data in SM. The description of G5K, G5H and G5HK has been moved in the Introduction before the new Chart 1 (lines 109-118). The text of the original manuscript has been consequently slightly modified.

  1. The sentences at line 96-98 concluding the toxicities, if newly discovered, should be moved to Results section. But Ref 21 was also cited here, does that mean the results in the Supplementary Materials and the Table S3 and S4 were reported in the Ref already? If so these results are not needed to be repeated in SM again (kind of self-plagiarism?)

Concerning the sentence at lines 96-98 (unrevised manuscript) reporting the absence of toxicity for G5Ds, the Reviewer is right in thinking that results from cytotoxicity essay have been already reported in Ref. 21 (Now Ref. 30 in the revised manuscript). Therefore, the sentence has been modified by removing references to Table S3 and S4 in SM and Section S2 in SM has been removed. Please see lines 143-145 in the revised manuscript.  

  1. Keep the references with same formats. For example, capital or don’t capital all the first letters at titles of the references.

All the references have been carefully checked and adjusted in the same format.

In addition, after having checked carefully the manuscript some improving changes have been made by the authors themselves.

Round 2

Reviewer 1 Report

The paper has been duly corrected. In the present form is suitanable for publication in Polymers

Reviewer 2 Report

The authors revised the manuscript according to the comments from the reviewers. it is ready to publish.